# Effects of Nitrogen and Phosphorus Regulation on Plant Type, Population Ecology and Sheath Blight of Hybrid Rice

**DOI:** 10.3390/plants11172306

**Published:** 2022-09-02

**Authors:** Guotao Yang, Rong Liu, Peng Ma, Hong Chen, Rongping Zhang, Xuechun Wang, Yongyan Li, Yungao Hu

**Affiliations:** 1College of Life Science and Engineering, Southwest University of Science and Technology, Mianyang 621010, China; 2Radiology Department, Mianyang Central Hospital, Mianyang 621000, China

**Keywords:** nitrogen and phosphorus, hybrid rice, population temperature and humidity, sheath blight

## Abstract

(1) Background: Sheath blight is one of the most economically significant rice diseases worldwide. A study was conducted in order to find the relationship and impact of the amount of nitrogen (N) and phosphorus (P) application on the hybrid rice population microclimate and the severity of sheath blight. (2) Methods: Four N and four P application levels were used to determine their impact on plant type, temperature, and humidity variation in different positions of population and the severity of sheath blight in the later stage. (3) Results: We found that N and P application levels could affect the plant type and change the population temperature and humidity by increasing the leaf length and leaf angle. (4) Conclusions: N application had a more significant (*p* < 0.05) impact on the plant type. High N application caused decreased temperature (hybrid rice population), while increased humidity (especially the population base layer at grain filling stage) resulted in severe sheath blight. High P application had similar impacts; however, P application increased material and nitrogen transport in plants and reduced the severity of sheath blight.

## 1. Introduction

Although paddy microclimates are formed under the background of macroclimate, they are different from macroclimate. A warm and humid climate is an essential part of the paddy microclimate, which significantly affects the yield and quality of hybrid rice [1,2]. However, temperature and humidity also enhance the growth environment for the development of infectious pathogens [3]. Previous studies found that the excessive N application increased rice plant height, stem tiller number, and decreased leaf area index (LAI). Such plant architecture enhances the permeability within the population, promotes CO_2_ diffusion, improves photosynthetic efficiency, and increases material accumulation [4]. However, the increase in LAI reduces permeability and changes the microclimate of the population.

Sheath blight is a highly infectious rice plant disease caused by the invasion of Rhizoctonia solani Kühn. This disease can easily grow, reproduce, and infect in high temperatures and humidity [5,6,7]. Therefore, it mainly occurs in the middle and late stages of rice plant growth and destroys the leaf sheath first. In this aspect of the molecular study, multiple genes are responsible for controlling plant infections, such as sheath blight. At present, there are no previous reports of significant disease resistance genes with complete host resistance [8,9,10]. Besides the microclimate, excessive N application significantly increased the incidence of false rice smut, rice blast, and sheath blight [11,12]. Increasing N application could significantly increase rice leaves’ nutrient content [13]. There was a close correlation between the N content of rice leaves and photosynthetic production capacity. In a specific range, the increase in N content in leaves was conducive to improving its single leaf photosynthetic rate, material accumulation, and yield. However, the increase in N content in leaves makes them more vulnerable to pathogen infection [8,10]. The application of trace elements, especially silicon, could significantly improve the activity of defense enzymes in the rice leaf sheath and increase the resistance to sheath blight [14,15].

There are few studies on the pathogenesis of rice sheath blight by P application. At the same time, there are no reports on the effects of N and P regulation on rice plant type, population microclimate, and the pathogenesis of sheath blight. Therefore, this experiment studied earlier situations and discussed their internal relationships. This experiment expects to adjust plant type and optimize the population microenvironment through N and P regulation to increase production or reduce the occurrence of diseases and pests.

## 2. Results

### 2.1. Effects of N and P Regulation on Rice Plant Morphology

N and P regulation significantly affected leaf length, leaf base angle, leaf opening angle, plant height, effective panicle numbers, and other plant type traits (Table 1). The plant height and effective panicles were mainly affected by N application. The length of D1 was affected by N more than P, and the length of D2 was more affected by P. At the same time, P has a more significant impact on leaf angle.

The leaf length increased significantly with N application. With the N dosage increase, the leaf length of Deyou 4727 decreased first and then increased, while the basal leaf angle decreased, and the leaf opening angles of D2 and D3 tended to increase; however, it showed little effect on the opening angle of the flag leaf. N application significantly increased plant height and effective panicles. In detail, plant height increased by 5.27% (N1), 8.27% (N2), and 15.49% (N3), respectively, and effective panicles increased by 4.07% (N1), 24.39% (N2), and 9.76% (N3), respectively. P application showed no significant difference under medium (P2) and high P (P3). It also significantly affected leaf angle, mainly increasing the basal angle and opening angle of leaves but had little effect on plant height and effective panicles, respectively.

The yield of Deyou 4727 increased significantly with the increase in N application, but the effect of P application on the yield was not significant (Figure 1). Under the condition of N0, the yield of Deyou 4727 was about 6000 kg/ha. After applying nitrogen fertilizer, the average yield of Deyou 4727 reached 9535.49 kg/ha, which was 15.85%, 38.70%, and 59.28% higher than that of N0.

### 2.2. Response of Temperature and Humidity to Rice Population and N Regulation

#### 2.2.1. Response of Population Temperature at Heading Stage to N Regulation

During the heading stage, the climatic conditions were relatively stable, and the population temperature was between 24–45 °C. The canopy temperature during the day was significantly lower than the external environment, but there was no significant difference at night. The base layer temperature during the day was generally higher than the external but opposite at night (Figure 2A,B). These findings suggest that the population temperature of rice was affected by the external environment, and there were also huge differences in the canopy and base layer temperature.

We compared and analyzed the daily variations in population temperature on 26 July (Figure 2C,D). We found that from 10:00 to 18:00, the canopy temperature of the population in N0 was significantly higher than that of other treatments. The population′s base layer temperature was slightly different under different nitrogen fertilizer treatments throughout the day. When the external temperature was relatively low (00:00–6:00 and 22:00–24:00), the base layer temperature of N0 was high, N3 was low, and the difference between N1 and N2 was not significant. However, in the period when the external temperature changed significantly (14:00–22:00), the order of base layer temperature was N0 > N1 > N2 > N3. These results show that the base layer temperature of the rice population decreases with the increase in N application.

#### 2.2.2. Response of Population Humidity to N Regulation at the Heading Stage

There was a negative relationship between population humidity and temperature at the heading stage (Figure 3A,B). The population humidity decreased with temperature increase during the day and increased with the decrease in temperature at night. The highest population humidity appeared between 0:00 and 6:00, the highest canopy humidity was 95%, and the highest base layer humidity was close to 100%. The minimum population humidity appeared between 12:00 and 15:00 and was generally lower than 60%. The variation range of canopy humidity with the external environment was more significant than that of the base layer. Except for some periods of N1 treatment, the population humidity was higher than the external humidity most of the time. This phenomenon also existed in the canopy at night and the base during the day. From 26 July to 30 July, the minimum population humidity increased gradually, negatively corresponding to the variation in population temperature.

After comparing the diurnal variation in population humidity on 26 July, we found that the variation law of heading date with time was the same. Under different N treatments, the daily variation in canopy and base layer humidity of the population decreased first and then increased (Figure 3C,D), and the base layer humidity was higher than that of the canopy. Different N treatments significantly impacted the population canopy humidity from 10:00 to 18:00. In this period, the average canopy humidity of different N treatments was 51.57%, 54.20%, 52.13%, and 54.46%, respectively, indicating that increasing N application can increase population canopy humidity. The effects of different N treatments on the base layer humidity of the population were significantly different throughout the day. However, the changes were relatively stable from 0:00 to 8:00 and 21:00 to 24:00. The humidity of N1 was the lowest, while N3 humidity was higher than N1 from 21:00 to 7:30 the next day, and it was higher than other treatments from 7:30 to 21:00. The variation trend of N2 and N4 was opposite to that of N3 treatment. The results showed that an optimum amount of N application could reduce the humidity at the base layer of the population at night and increase it during the day.

#### 2.2.3. Response of Population Temperature to N Regulation at Grain Filling Stage

We analyzed the response of rice population temperature to N regulation at the grain filling stage (Figure 4). The population temperature was lower than the external environment. The difference between the canopy and external temperature was smaller than at the base layer. There were also differences among different N treatments. There were no significant differences in the temperature between the canopy and the external environment at night, but there were significant differences in the daytime. Compared with the canopy, the temperature at the base layer of the population was more different from the external temperature. The complete emergence of rice panicles increased the canopy density. We found that when the external temperature decreased at night, the population’s base layer’s temperature slightly increased compared to the external and canopy temperature. These results indicate that compared with the canopy, the temperature at the base layer of the population at the grain filling stage was lower than in the external and could delay reducing the external temperature at night.

We compared the daily variation in population temperature on 20 August (Figure 4). From 10:00 to 18:00, the canopy temperature reached above 40 °C, and the base layer temperature was lower than 40 °C. Different N treatments had significant effects on the maximum temperature of the population. The canopy temperature in the daytime was the highest in N0, followed by N2, then N1, and the lowest was found in N3. The population base layer temperature was N1, N0, N2, and N3. These results indicate that the N application could reduce the population temperature, and the reduction effect of population temperature was the strongest under the condition of high N.

#### 2.2.4. Response of Population Humidity to N Regulation at Grain Filling Stage

The negative relationship between population humidity and temperature at the grain filling stage decreased significantly with increasing temperature (Figure 5A,B). At the grain filling stage, low population humidity was found from 12:00 to 15:00. In the five selected days of the grain filling stage, the population humidity differed from the external environment and was more significant than the external. The change in canopy humidity with the external environment was more significant than that of the base layer. The base layer humidity of the population was significantly higher than that of the canopy in the daytime.

We compared the daily variation in population humidity on 20 August (Figure 5C,D). We found that in the grain filling stage, the minimum humidity of the canopy was 30% to 45%, the base layer minimum humidity was 45% to 60%, and the maximum humidity was close to 100%. The population humidity had different responses to each N treatment, especially in the daytime. The humidity response to N treatment in different parts of the rice population was significantly different. The difference in canopy humidity in different N treatment populations was slight, but base layer humidity was significant, and the maximum base layer humidity was also quite different. From 10:00 to 18:00, there was a significant difference in canopy humidity, which was N3 > N0 > N2 > N1, and the base layer humidity was N2 > N0 > N3 > N1. The variation range of population humidity for N3 was the lowest.

### 2.3. Response of Temperature and Humidity of Rice Population to P Regulation

#### 2.3.1. Response of Population Temperature at Heading Stage to P Regulation

Under different P application conditions, the variation law of population temperature was the same as the external. When the external temperature changed rapidly, the population canopy temperature changed quickly, while the base layer changed slowly (Figure 6A,B). The lowest population temperatures for the base layer were lower than the external, and the highest temperature was higher than the external environment. The difference between different P treatments was smaller than that of N.

This experiment used the daily variation in population temperature on 26 July for analysis (Figure 6C,D). The change law of population temperature at different times of the day was similar to that of N treatment. There was little difference in the canopy temperature of the population with increased P application. The base layer temperature of other treatments was lower than P0 before reaching the maximum temperature and higher than P0 after reaching the maximum temperature.

#### 2.3.2. Response of Population Humidity to P Regulation at the Heading Stage

The population’s upper and lower humidity increased with the increase in P application at the heading stage. The gradient of humidity difference between medium P (P2) and high P (P3) treatments was slight, and this law also existed between no P (P0) and low P (P1) treatments. The population canopy humidity of P2 and P3 treatment was significantly different from P0 and P1 treatment. The humidity gradient at the population base was uniform under different phosphorus treatments (Figure 7A,B).

We also analyzed the daily variation in population humidity on 26 July (Figure 7C,D). The canopy humidity decreased significantly after 6:00, decreased to less than 70% around 9:00, and increased to more than 70% around 19:00. From 11:00 to 18:00, the population canopy humidity changed between 50% and dropped below 50% from 13:00 to 14:00. The maximum population humidity appeared between 0:00 and 6:00, and the maximum humidity of the canopy and base layer was close to 100%. The variation law of the population base layer humidity was similar to the canopy, but the maximum humidity was higher than that of the canopy. Compared with the canopy, the base layer humidity of the population had a more substantial buffering effect on the external environment. Under different P treatments, the difference in canopy humidity was P3 > P2 > P1 > P0. In the daytime, high P treatment (P3) significantly increased the minimum humidity of the canopy.

#### 2.3.3. Response of Population Temperature to P Regulation at Grain Filling Stage

Figure 8 shows that the population canopy temperature was higher than the base layer at the grain filling stage. Under different P treatments, the maximum temperature of the population was low compared to the external environment. The minimum temperature at the population base was higher than the external environment.

On 20 August, we used the daily variation in population temperature for analysis (Figure 8C,D). We found that population temperature increased significantly at 7:00; the canopy temperature reached more than 30 °C at 9:00 and did not drop below 30 °C until 20:00. From 10:00 to 19:00, the population temperature changed between 35 °C and from 13:00 to 17:00, the canopy temperature was above 40 °C. The variation law of population base layer temperature was similar to that of the canopy, but the maximum temperature was lower than that of the canopy. Different P treatments had a low significant difference in population base layer temperature, but there was a significant difference in the canopy. The population canopy temperature at night was P3 > P2 > P1 > P0, the opposite during the daytime. The higher the P amount, the variation in canopy temperature, the more gentle, showing a relatively significant buffer effect on the variation for external.

#### 2.3.4. Response of Population Humidity to P Regulation at Grain Filling Stage

At the grain filling stage, under different P application conditions, the population humidity was mainly higher than the external, and the population baselayer minimum humidity was higher than the canopy. With the increase in P application, the population’s highest and lowest humidity showed an increasing trend (Figure 9).

This experiment used the daily variation in population humidity on 20 August for analysis (Figure 9C,D). The canopy humidity significantly decreased after 7:00, reached below 70% after 9:00, and increased to more than 70% after 19:00. From 11:00 to 18:00, the population humidity changed between 50%, and from 12:00 to 17:00, the canopy humidity dropped to about 40%. The maximum population humidity appeared between 0:00 to 6:00, and the maximum humidity of the canopy and base layer was close to 100%. The variation law of baselayer humidity was similar to that of the canopy. However, it decreased to less than 70% after 10:00 and increased to more than 70% after 18:00. Moreover, the minimum humidity was higher than that of the population canopy.

High P treatment (P3) could significantly increase the minimum canopy humidity. At night, the variation in population humidity with P application was similar to that of the canopy, and the difference between different P treatments was relatively small. Under different P treatments, the difference in base layer humidity during the daytime was low compared to the night. There was no significant difference in other treatments except that low P treatment (P1) significantly increased base layer humidity in the daytime.

### 2.4. Effects of N and P Regulation on the Severity of Rice Sheath Blight

The severity of rice sheath blight was significantly affected by the regulation of N and P application. We found that N application significantly increased the occurrence of sheath blight, and P application significantly inhibited the occurrence of sheath blight (Table 2). The severity of sheath blight increased significantly with the increase in N application. Under the condition of N application, the severity of sheath blight increased by 159.53% (N1), 413.49% (N2), and 822.58% (N3), respectively, compared with no N treatment (N0). Under the condition of phosphorus application, compared with the treatment without N (N0), the incidence severity of sheath blight decreased by 17.21% (P1), 38.07% (P2), and 59.96% (P3), respectively. It proved that the effect of P on rice sheath blight was lower than that of N.

### 2.5. Correlation Analysis of N and P Regulation, Population Temperature, Humidity Change, and Sheath Blight

#### 2.5.1. Correlation Analysis of N Regulation, Population Temperature, Humidity Variation, and Sheath Blight

In order to analyze the correlation between N application rate, the variation in population temperature and humidity, and the severity of sheath blight, the experiment used five days at the heading stage and grain filling stage, respectively, to calculate the average values of temperature and humidity. Then, the average values, N application rate, and sheath blight severity were analyzed (Table 3). There was a significant positive correlation between the severity of sheath blight and the amount of N application and a significant negative correlation with the daily average humidity at the population base. The population canopy and base temperature and humidity have a significant correlation. N application greatly influenced the daily average temperature and humidity at the population base. It has a positive correlation with base temperature and a negative correlation with population humidity. In the filling stage, the daily average humidity at the population base reached a significant level.

#### 2.5.2. Correlation Analysis of P Regulation, Population Temperature, Humidity Change, and Sheath Blight

The experiment analyzed the correlation between P application rate, population temperature, humidity variation, and sheath blight severity. The P application rate was negatively correlated with the severity of sheath blight and positively correlated with the daily average temperature of the population (Table 4). The severity of sheath blight was significantly correlated with the daily average temperature of the population canopy, positively correlated with the daily average humidity of the population canopy, and not significantly correlated with the daily average humidity of the population base layer.

## 3. Discussion

### 3.1. N and P Regulation Significantly Affected Hybrid Rice Plant Morphology

In addition to being controlled by genes, rice plant traits such as plant height and tillering were also affected by fertilizers, exogenous regulators, cultivation measures, environmental factors, and other factors [16,17,18]. The application of N and P affected the establishment of rice plant types in many ways, and the plant type had a significant impact on population temperature and humidity [19,20]. N application increased plant height, leaf area, and tiller number. The higher the tiller number, the larger the leaf area, the larger the light-receiving area, and the higher the light energy utilization rate [21]. The application of P promoted plant growth and root development, promoted the transfer of nutrients to grains, and increased the grain number and weight in the later stage of growth to increase 1000 grain weight and yield [22]. This experiment found that the application of N and P significantly affected the plant traits of rice. N application increased plant height and panicle number and promoted the increase in leaf length but had no significant effect on leaf width. With the increase in leaf length, leaf biomass increased, leaf base angle and leaf opening angle increased, and the leaves appeared bent. At present, most of the flag leaves of hybrid rice varieties are straight, so N application had little effect on the leaf angle of flag leaves, mainly on D2 and D3 leaf angles. The P application also increased the length of rice leaves and increased the leaf angle.

### 3.2. Rice Plant Type Variation Caused Population Microclimate Difference

Crop type and population structure significantly affect population microclimate variation. Prasad found significant differences in canopy temperature among different rice varieties [23]. Fischer’s study found that wheat varieties with different genotypes or different fertilization treatments of the same variety have different population structures, resulting in different inter and intra temperatures. The temperature and humidity variation caused by crop population structure could further affect the variation in plant type and yield components [24]. Garrity found a significant negative correlation between canopy temperature and seed setting rate under an abnormally high-temperature environment [25]. At the same time, variation in population humidity could cause variation in leaf stomatal conductance, transpiration, yield, and quality [26].

This study found that the temperature of the rice population was lower than the external environment, the humidity was higher than the external, and there was a significant correlation between temperature and humidity. The difference between population temperature and humidity at the heading stage and the external environment was less than that at the grain filling stage, which may be due to the better erect panicle at the heading stage and the rapid ventilation between the canopy and the external environment. The temperature difference between the canopy and the external rice population was less between the base and external layers. The increase in N application increased the leaf length and leaf angle, resulting in the bending of D2 and D3. Therefore, the light transmittance in the middle and base layer of the population decreased, the solar radiation received by the base layer decreased, and respiration increased. The effect of nitrogen fertilizer on panicle number and leaf was relatively significant, so the difference in population temperature and humidity between different N treatments was more significant than that of P treatment.

### 3.3. Population Temperature and Humidity Variation Was an Important Factor Affecting the Incidence of Rice Sheath Blight

Rice sheath blight disease has increased due to intensifying rice production and increased chemical fertilizer use. Sheath blight has become one of China’s three major rice diseases [27], posing a significant threat to China’s food security. The quality of rice infected with sheath blight is significantly decreased [28]. Rice blast reduces internode folding resistance and increases the risk of rice lodging [29]. They found that the occurrence degree of diseases and pests such as rice blasts and sheath blight are closely related to the microclimate of the paddy field [12,30]. Excessive N application reduced population permeability, reduced the population base layer temperature, significantly increased humidity, and increased the infection probability of pathogenic bacteria among plants [31].

On the other hand, excessive N application reduced the degree of silicification of epidermal plant cells and increased the content of amino acids [32]. All these provide favorable conditions for the infection and occurrence of rice blasts and other diseases [33,34]. A higher N application rate increases population density and causes disease. Rational N application at the booting and flowering stages and planting space optimization were the best choices for controlling rice sheath blight [35]. In this experiment, the increased application of N fertilizer significantly increased the severity of rice sheath blight. The application of P fertilizer had a specific inhibitory effect on the incidence of sheath blight. The severity of rice sheath blight was mainly related to the population temperature and humidity at the grain filling stage, especially the variation in temperature and humidity at the population base layer.

In conclusion, the application of nitrogen and phosphorus significantly affects the plant type of rice. The application of nitrogen fertilizer significantly increases the length of the leaves, while the application of phosphorus fertilizer mainly affects the angle of the leaves. At the same time, the application of nitrogen fertilizer also causes a significant increase in plant height and effective panicles. The increase in biomass and effective panicle under the condition of increasing nitrogen fertilizer is the main reason for the increase in yield. In this experiment, increased N application significantly increased the severity of sheath blight, and increased P application improved the resistance of rice plants to sheath blight. The main reason is that the application of N and P changes the rice plant type and then changes the population temperature and humidity. N application increased the severity of sheath blight, which caused the deterioration of the population microclimate and physiological changes in plants under the N application. Although the P application also leads to the deterioration of the population microclimate, P fertilizer can promote rice plants’ material and N output. Under the condition of P application, the high material output at the population base layer may be the main reason for the excellent permeability in the later stage of the population and for reducing the occurrence of sheath blight.

## 4. Materials and Methods

### 4.1. Experimental Design

Mianyang is located in the northwest of the Sichuan Basin, in the middle and upper reaches of the Fujiang River, with mountains accounting for 61.0%, hills accounting for 20.4%, and plain accounting for 18.6%. The annual average temperature is 14.7–17.3 °C, and the frost-free period is 252–300 days. The average annual precipitation is 826–1417 mm, and the annual sunshine is about 1300 h.

The experiment was a long-term positioning experiment. The contents of soil total N, P, and potassium (K) were 143.6 mg/kg, 68.5 mg/kg, and 66.4 mg/kg, respectively, and the available N, P, and K were 78.8 mg/kg, 50.3 mg/kg, and 68.5 mg/kg, respectively. The experiment designed the N application as N0, N1, N2, and N3 (the pure N application rates were 0 kg/ha, 120 kg/ha, 180 kg/ha, and 240 kg/ha, respectively). The experiment designed the P application as P0, P1, P2, and P3 (the P_2_O_5_ application rates were 0 kg/ha, 60 kg/ha, 120 kg/ha, and 180 kg/ha, respectively). K fertilizer was K_2_O, and the application rate was 120 kg/ha. The fertilizer operation research method applied N as the base, tiller, and panicle fertilizer in the ratio of 5:3:2 and applied P only as the base fertilizer. K was applied as a base and panicle fertilizer in the ratio of 1:1. Cross design, seven treatments were N0P1, N1P1, N2P1, N3P1, N1P0, N1P2, and N1P3. To study the effects of different nitrogen and phosphorus fertilizer treatments on the microclimate of the rice population and the incidence of sheath blight (N1 and P1 are the normal application rates of local rice fields, that is, to study the effects of four nitrogen fertilizer treatments under normal phosphorus application conditions or four phosphorus fertilizer treatments under normal nitrogen application conditions on the microclimate of rice population and the incidence of sheath blight). The test plot was 20 m long and 7 m wide. Each test section was separated with a 50 cm high and 30 cm thick cement ridge to prevent fertilizer and water. In 2016, we positioned the N and P treatment in the test field.

We used Deyou 4727 as the test rice variety. It was sown on 10 April 10 and transplanted on 12 May 2019. The planting specification was 33.3 cm × 16.7 cm. Each population consisted of 15 rows and 15 plants in each row, repeated 3 times.

### 4.2. Sampled and Determination

#### 4.2.1. Determination of Plant Type Characteristics

At the full heading stage, five representative rice plants were selected with the five-spot-sampling method, and the leaf length, basal leaf angle (including the angle between stem and leaf base), leaf opening angle (including the angle between the stem and connecting line from leaf base to tip) of the flag leaf (D1), inverted second leaf (D2: from the sword leaf down to the second leaf), and inverted third leaf (D3: from the sword leaf down to the third leaf), plant height, and effective panicles were measured.

#### 4.2.2. Measurement of the Hybrid Rice Population Temperature and Humidity

After turning green, X automatic temperature and humidity recorders were installed in the canopy (middle of panicle layer) and the base layer (1/4 plant height) of the rice population; we recorded the data every half hour, and then the data were transmitted to the data center through a wireless transmission system for storage. The heading stage (26–30 July) and grain filling stage (16–20 August) on continuous breezy and sunny days were selected to analyze the temperature and humidity changes inside and outside the rice population after N and K regulation treatment. At the same time, three days with similar external weather were selected to compare the population temperature and humidity and analyze its response to N and P regulation.

#### 4.2.3. Investigation of Sheath Blight in the Field

According to GBT15791-2011 *Specification for Forecasting and Investigation of Rice Sheath Blight*, 20 days after the full heading stage, 30 plants for each treatment were selected to investigate the severity of sheath blight. See Table 5 for severity classification. Severity is calculated by Formula (1).
(1)$$A=\frac{\sum \left(Bi\times Bd\right)}{M\times Md}\times 100$$

*A*: severity

*B_i_*: number of disease plants with different levels of severity

*B_d_*: representative value of severity at all levels

*M*: total number of plants surveyed

*M_d_*: highest representative value of severity

#### 4.2.4. Data Analysis and Plotting

We used Excel 2017 for data statistics and sorting, SPSS 19.0 to analyze variance and compare difference significance, and Origin 9.0 for plotting. Each treatment value is expressed as mean ± standard deviation (SD) of 3–6 biological replicates. The differences were determined by Student’s *t*-test at *p* < 0.05 (*), 0.01 (**), or 0.001 (***).

## Figures and Tables

**Figure 1 plants-11-02306-f001:**
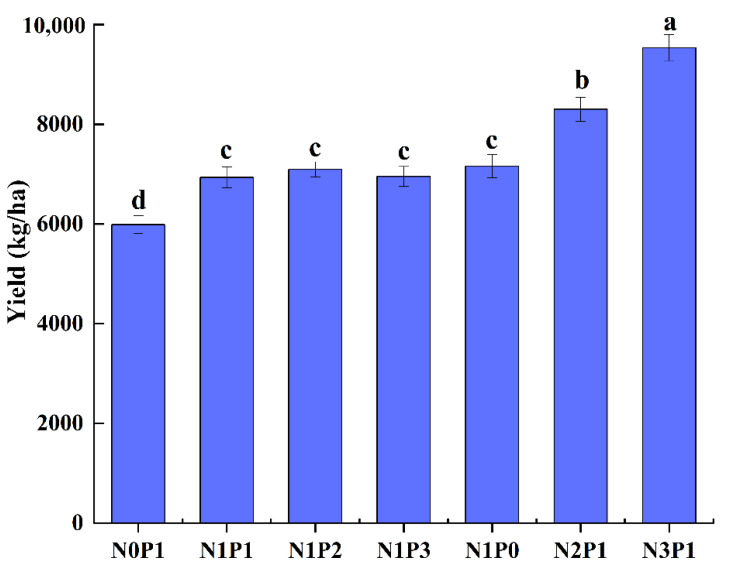
Effect of nitrogen and phosphorus regulation on yield of hybrid rice. The values represent the mean of three replicates. Different letters indicate a significant difference at *p* < 0.05.

**Figure 2 plants-11-02306-f002:**
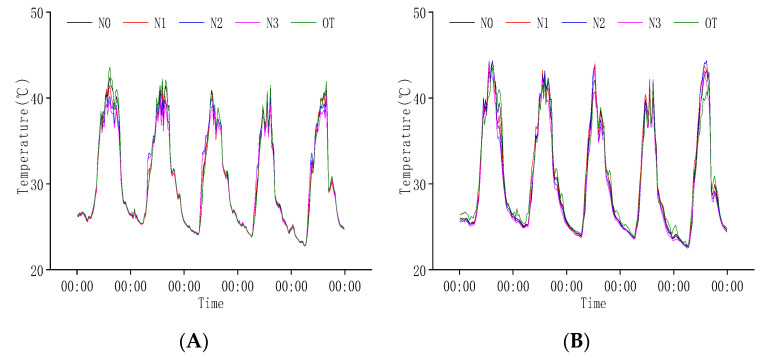
Variation in population temperature at heading stage. (**A**) Canopy population temperature from 26 July to 30 July. (**B**) Baselayer temperature of the population from 26 July to 30 July. (**C**) Canopy temperature of the population on 26 July. (**D**) Baselayer temperature of the population on 26 July. OT is the change in external temperature, N0, N1, N2, and N3 are the effects of different nitrogen fertilizers on population temperature under P1 conditions.

**Figure 3 plants-11-02306-f003:**
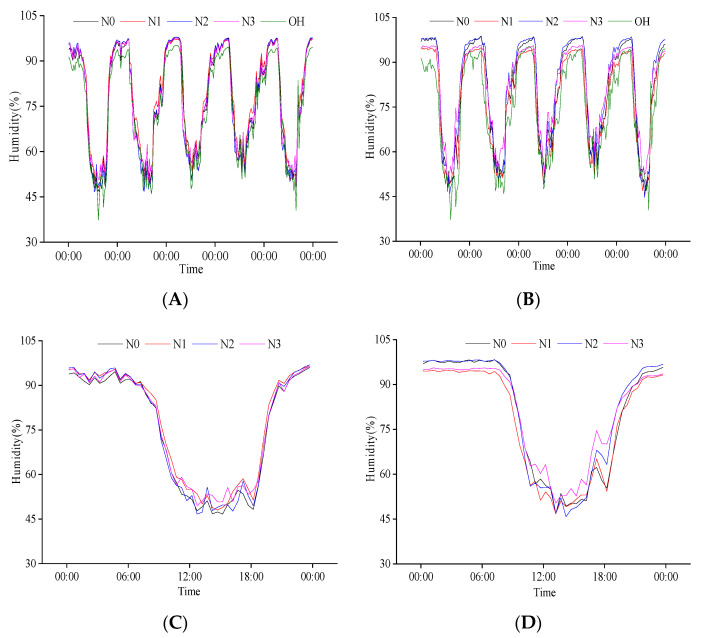
Daily variation in population humidity at the heading stage. (**A**) Canopy humidity of population from 26 July to 30 July; (**B**) Baselayer humidity of population from 26 July to 30 July; (**C**) Canopy humidity of population on 26 July; (**D**) Baselayer humidity of population on 26 July. OH is the change in external humidity, N0, N1, N2, and N3 are the effects of different nitrogen fertilizers on population humidity under P1 conditions.

**Figure 4 plants-11-02306-f004:**
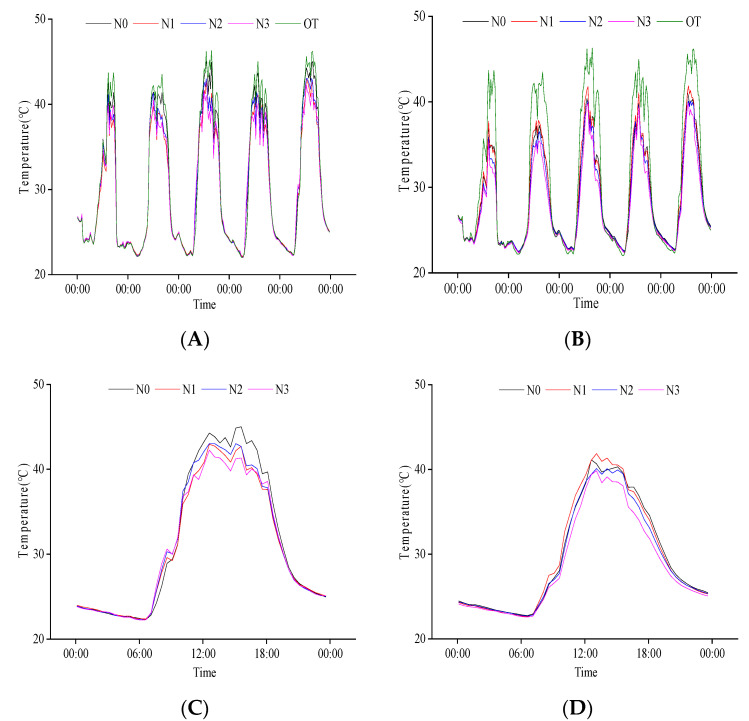
Daily variation in temperature at grain filling stage. (**A**) Canopy population temperature from 16 August to 20 August; (**B**) Baselayer temperature of the population from 16 August to 20 August; (**C**) Canopy temperature of the population on 20 August; (**D**) Baselayer temperature of the population on 20 August. OT is the change in external temperature, N0, N1, N2, and N3 are the effects of different nitrogen fertilizers on population temperature under P1 conditions.

**Figure 5 plants-11-02306-f005:**
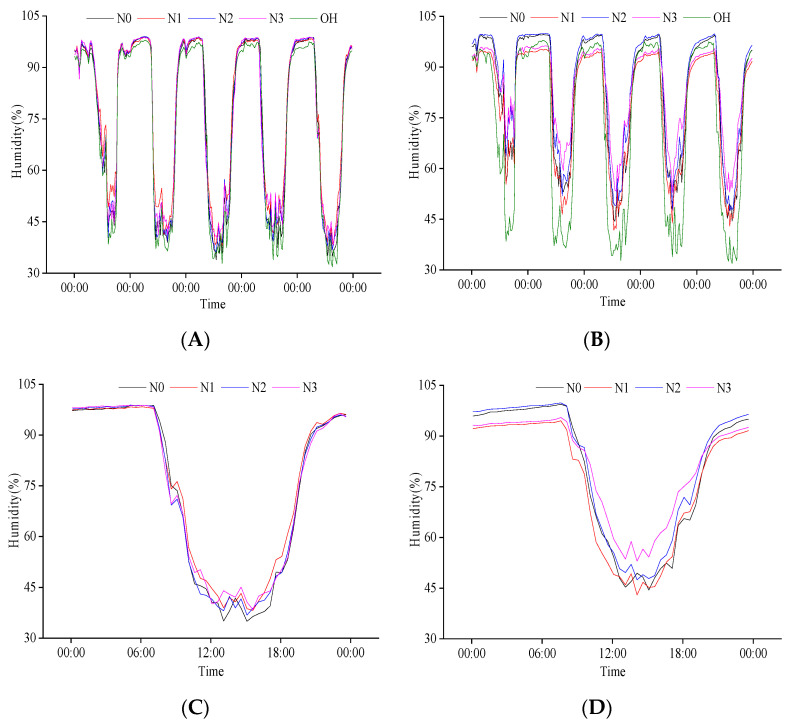
Daily variation in population humidity at grain filling stage. (**A**) Canopy humidity of population from 16 August to 20 August; (**B**) Baselayer humidity of population from 16 August to 20 August; (**C**) Canopy humidity of population on 20 August; (**D**) Baselayer humidity of population on 20 August. OH is the change in external humidity, N0, N1, N2, and N3 are the effects of different nitrogen fertilizers on population humidity under P1 conditions.

**Figure 6 plants-11-02306-f006:**
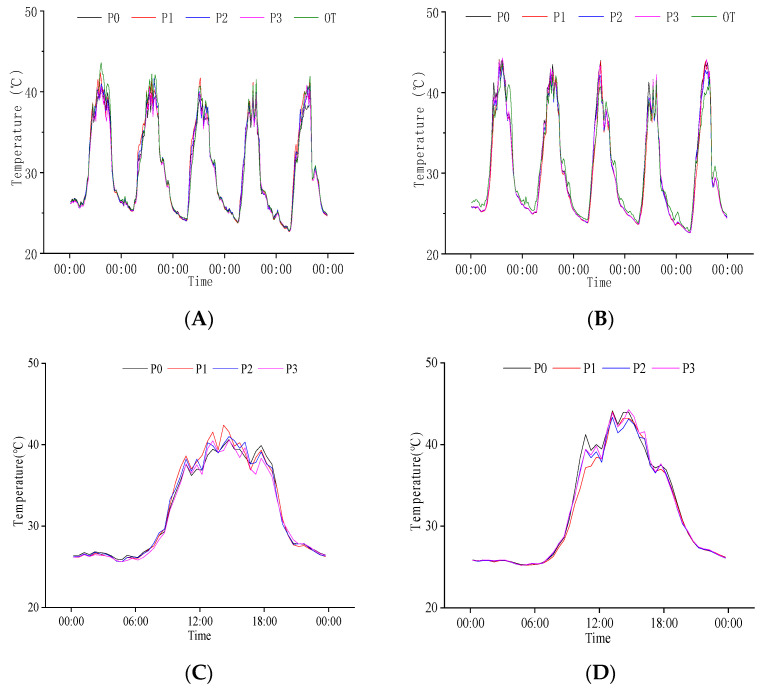
Variation in population temperature at heading stage. (**A**) Canopy population temperature from 26 July to 30 July; (**B**) Baselayer temperature of the population from 26 July to 30 July; (**C**) Canopy temperature of the population on 26 July; (**D**) Baselayer temperature of the population on 26 July. OT is the change in external temperature, P0, P1, P2, and P3 are the effects of different nitrogen fertilizers on population temperature under N1 conditions.

**Figure 7 plants-11-02306-f007:**
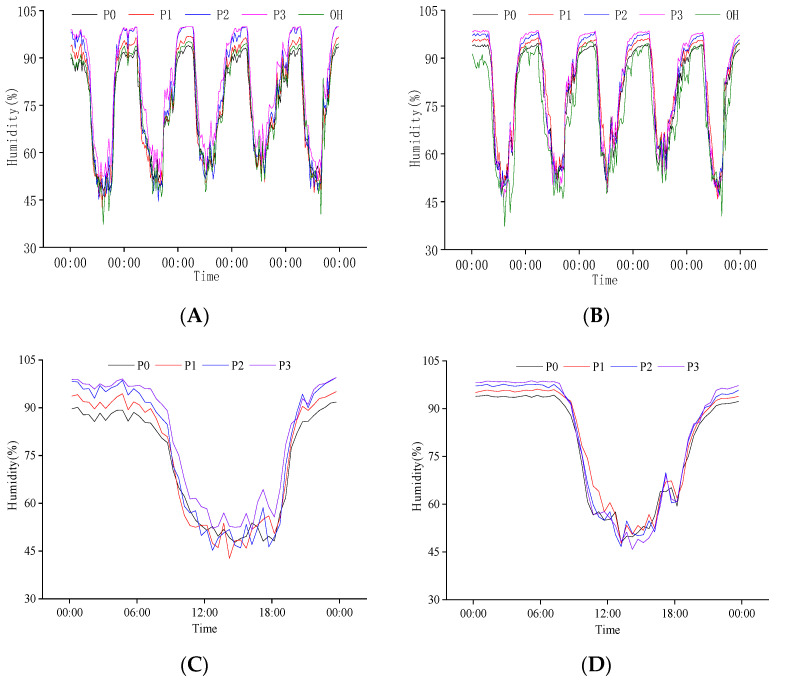
Daily variation in population humidity at the heading stage. (**A**) Canopy humidity of population from 26 July to 30 July; (**B**) Baselayer humidity of population from 26 July to 30 July; (**C**) Canopy humidity of population on 26 July; (**D**) Baselayer humidity of population on 26 July. OH is the change in external humidity, P0, P1, P2, and P3 are the effects of different nitrogen fertilizers on population humidity under N1 conditions.

**Figure 8 plants-11-02306-f008:**
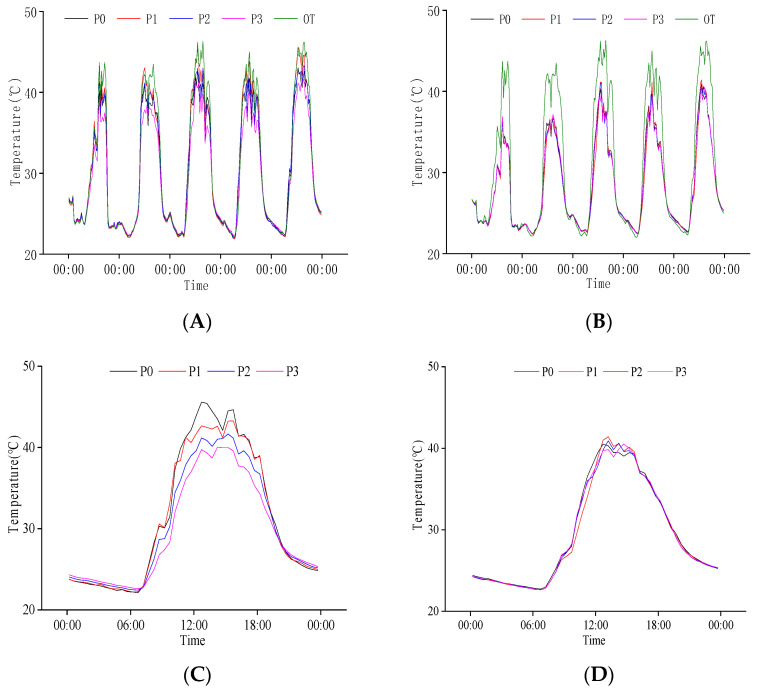
Daily variation in population temperature at the grain filling stage. (**A**) Canopy humidity of population from 16 August to 20 August; (**B**) Baselayer humidity of population from 16 August to 20 August; (**C**) Canopy humidity of population on 20 August; (**D**) Baselayer humidity of population on 20 August. OT is the change in external temperature, P0, P1, P2, and P3 are the effects of different nitrogen fertilizers on population temperature under N1 conditions.

**Figure 9 plants-11-02306-f009:**
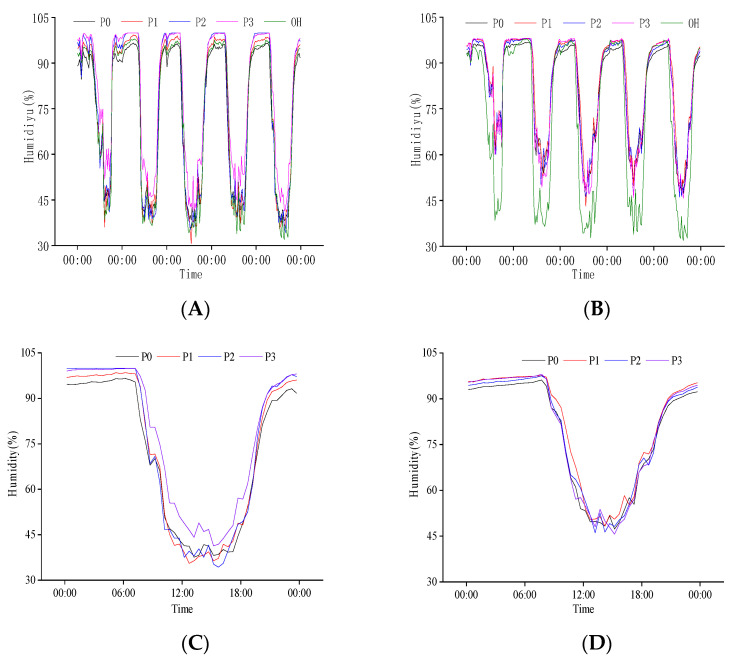
Daily variation in population humidity at grain filling stage. (**A**) Canopy humidity of population from 16 August to 20 August; (**B**) Baselayer humidity of population from 16 August to 20 August; (**C**) Canopy humidity of population on 20 August; (**D**) Baselayer humidity of population on 20 August. OH is the change in external humidity, P0, P1, P2, and P3 are the effects of different nitrogen fertilizers on population humidity under N1 conditions.

**Table 1 plants-11-02306-t001:** Effects of nitrogen and N regulation on rice plant morphology.

	Leaf Length (cm)			Leaf Basal Angle (°)			Leaf Opening Angle (°)			Plant Height (cm)	Effective Panicle (×10^3^/ha)
	D1	D2	D3	D1	D2	D3	D1	D2	D3		
N0P1	22.00 c	36.83 d	47.23 c	17.77 a	36.77 d	51.30 cd	17.50 ab	40.43 b	45.07 e	110.83 d	1845.00 c
N1P1	29.93 a	41.40 b	51.53 b	15.50 b	41.37 c	49.53 d	18.47 a	37.30 c	49.40 d	116.67 bc	1920.00 bc
N1P2	30.30 a	44.10 ab	56.33 a	15.53 b	44.07 b	49.83 d	18.73 a	46.43 a	53.73 c	118.67 b	1785.00 c
N1P3	29.23 a	47.23 a	58.20 a	19.33 a	47.13 a	54.47 b	15.50 b	37.43 c	59.50 a	114.33 cd	1995.00 bc
N1P0	23.97 b	34.97 d	52.03 b	15.43 b	35.47 d	60.13 a	12.03 c	46.37 a	48.53 d	114.00 cd	2010.00 ab
N2P1	22.20 c	37.57 c	49.50 bc	13.27 b	41.40 c	50.53 cd	16.60 ab	39.30 bc	53.30 c	120.00 b	2295.00 a
N3P1	29.50 a	43.43 b	56.37 a	12.33 b	43.53 b	52.27 bc	16.70 ab	46.70 a	56.77 b	128.00 a	2025.00 ab
N	58.63 **	22.04 **	28.99 **	28.42 **	49.87 **	3.50 ns	2.90 ns	37.80 **	57.36 **	59.64 **	6.66 **
P	26.53 **	60.86 **	20.38 **	17.81 **	117.26 **	64.38 **	37.84 **	62.48 **	56.83 **	5.54 *	1.82 ns

The values represent the mean of three replicates. Different letters indicate a significant difference at *p* < 0.05. ns: not significant (*p* > 0.05); **: significant at *p* < 0.01; *: significant at *p* < 0.05.

**Table 2 plants-11-02306-t002:** Effects of N and P regulation on the severity of rice sheath blight.

	P0	P1	P2	P3
N0	—	3.41 ± 0.67 e	—	—
N1	10.69 ± 2.14 c	8.85 ± 1.10 cd	6.62 ± 0.54 de	4.28 ± 1.19 e
N2	—	17.51 ± 1.00 b	—	—
N3	—	31.46 ± 2.22 a	—	—

The values represent the mean of three replicates. Different letters indicate a significant difference at *p* < 0.05.

**Table 3 plants-11-02306-t003:** Correlation Analysis among N application rate, population temperature, humidity, and sheath blight.

Stages		N	Canopy Temperature	Baselayer Temperature	Canopy Humidity	Baselayer Humidity
Heading stage	N	1.00				
	Canopy temperature	−0.47 ns	1.00			
	Base layer temperature	0.52 ns	−0.96 *	1.00		
	Canopy humidity	0.11 ns	−0.81 *	0.63 ns	1.00	
	Base layer humidity	−0.76 ns	0.57 ns	−0.75 ns	0.01 ns	1.00
	Severity of diseases	0.99 **	−0.56 ns	0.64 ns	0.15 ns	−0.84 *
Grain filling stage	N	1.00				
	Canopy temperature	0.26 ns	1.00			
	Base layer temperature	0.58 ns	−0.63	1.00		
	Canopy humidity	−0.74 ns	−0.84 *	0.11 ns	1.00	
	Base layer humidity	−0.88 *	0.09 ns	−0.79 ns	0.41 ns	1.00
	Severity of diseases	0.99 **	0.12 ns	0.69 ns	−0.64 ns	−0.94 *

The values represent the mean of three replicates. ns: not significant (*p* > 0.05); **: significant at *p* < 0.01; *: significant at *p* < 0.05.

**Table 4 plants-11-02306-t004:** Correlation Analysis among P application rate, population temperature, humidity, and sheath blight.

Stages		P	Canopy Temperature	Baselayer Temperature	Canopy Humidity	Baselayer Humidity
Heading stage	P	1.00				
	Canopy temperature	0.99 **	1.00			
	Base layer temperature	0.78 ns	0.72 ns	1.00		
	Canopy humidity	−0.37 ns	−0.45 ns	0.29 ns	1.00	
	Base layer humidity	0.05 ns	0.14 ns	−0.58 ns	−0.94 *	1.00
	Severity of diseases	−0.81 *	−0.81 *	−0.46 ns	0.55 ns	−0.25 ns
Grain filling stage	P	1.00				
	Canopy temperature	0.99 **	1.00			
	Base layer temperature	−0.05 ns	−0.11 ns	1.00		
	Canopy humidity	−0.83 *	−0.89 *	0.53 ns	1.00	
	Base layer humidity	−0.19 ns	−0.10 ns	−0.94 *	−0.36 ns	1.00
	Severity of diseases	−0.81 *	−0.90 *	0.20 ns	0.91 *	−0.12 ns

The values represent the mean of three replicates. ns: not significant (*p* > 0.05); **: significant at *p* < 0.01; *: significant at *p* < 0.05.

**Table 5 plants-11-02306-t005:** Classification criteria for severity of rice sheath blight.

Grade	Severity Classification Criteria
Grade 0	The whole plant is disease-free
Grade 1	The disease starts from basal leaf and sheath
Grade 2	The disease starts from the leaf sheath or leaf below the third leaf (from the top leaf, the same below)
Grade 3	The disease starts from the leaf sheath or leaf below the second leaf
Grade 4	Parietal leaf sheath or parietal leaf disease
Grade 5	Whole plant disease and wither

## Data Availability

Not applicable.

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
