# Peer review of "Effects of Nitrogen and Phosphorus Regulation on Plant Type, Population Ecology and Sheath Blight of Hybrid Rice"

_plants, 2022, doi:10.3390/plants11172306_

Round 1
Reviewer 1 Report
See attached document.

Author Response
Response to Reviewer 1
Comments and Suggestions for Authors
Dear Authors,
Understanding of rice plants response to fertilizer (like N and P) application is very important in agricultural practice. This work tries to study the plant phenotype, microclimate and blight sheath after different levels of N and P application. Please find the below points to improve your manuscript (major concerns in bold):
We want to thank you for the kind help and efforts that you invested in reviewing our manuscript and helping us to improve it. We believe that having incorporated your proficient suggestions, our manuscript is now improved in quality, comprehensiveness and clarity and that you will find it suitable for publication in Plants. We have addressed the point-by-point response to the reviewer within the text below.
Q1. L60: Maybe it is good to change “plant type” to “plant morphology, or plant phenotype”? leaf angle and plant height are morphological phenotypes.
A1. Thank you for this suggestion. We revised from “plant type” to “plant morphology” (L62).
Q2. L66: typo of table 1 title
A2. Thank you for this comment. We improved table 1 title (L69).
Q3. L275-284: incidence and severity are different. You may find high occurrence of blight late, but is not severe. So please clarify this.
A3. Thank you for this question. We rechecked and revised it (L294-306).
Q4. L374-375: should it be a reference here?
A4. Thank you for this suggestion. We added some references (L392-396).
Q5. L398-415: In this section, there is no description of field experiment site (location, latitude and longitude, etc). Can’t really understanding L411 and L412. 33.3 x 16.7 is this the space of planting?
A5. Thank you for this comment. We added the description of the field experiment site and (L424-488, and 448-450).
Q6. Are those different treatments (N0P1, N1P1, N2P1, N3P1, N1P0, N1P2, and N1P3) randomly allocated in the field? In total, a figure of the experiment design will be helpful for readers to understand.
A6. Thank you for this suggestion. We added information about these different treatments according to journal requirements we already added many figures in our manuscript that is why we cannot add any new figure (L437-445).
Q7. Since you have data on effective panicle number, would be helpful if the yield data can be provided.
A7. Thank you for this valuable comment. We added yield data (L80-86).
Q8. L408: why compare N effects at the Phosphorus level of P1? Similarly, compare P effects at N1? Because P1 and N1 are optimum rate in farming practice?
A8. Thank you for this question. As we already know that N1 and P1 used for farming practice that’s why we used different N and P doses along with these optimum rates because we want to know about the actual dose of N and P.
Q9. L421: what are inverted second leaf and inverted third leaf? Are they just second and third leaf counted from the top? Again, I would recommend a graph to explain rice plant phenotype.
A9. Thank you for this comment. We revised it and added detailed information about second and third leaf, and we already added many figures in our manuscript that is why we cannot add any new figure (L456-458).
Q10. L423-432: Since microclimate measurements (temperature and humidity) are import, I am wondering how many measurements for each treatment?
A10. Thank you for this question. We have added detailed information and Table 5 (L470-481)
Q11. L424: after turning green? maybe how long after transplanting will be better?
A11. Thank you for this comment. When it is turning green 7 days after transplanting is better.
Q12. Table 3: please change the first row (name of the measurements)
A12. Thank you for this suggestion. We added the name of measurements in Tables 3 and 4.
Q13. For Figure 1-8: please explain what are “OT and OH”. Meanwhile, please clarify it is the response to Nitrogen or Phosphorus. In addition, there is no statistical analysis for the temperature and humidity data. If you have replicates, would it be good to show proper statistics together with your daily patterns.
A13. Thank you for this valuable comment. We revised and added information about OT and OH in all figure footnotes.
Q14. Reference 28 is about plant response to N in indica rice. Could you compare the results between this with your hybrid rice?
A14. Thank you for this comment. Yes, we compare plant response to N in indica rice with our hybrid rice.
Q15. A section of conclusion will be needed after discussion.
A15. Thank you for this suggestion. We added a conclusion section after the discussion.
Q16. English language of this manuscript should be improved significantly.
A16. Thank you for this suggestion. We have revised the whole manuscript carefully and tried to avoid any grammar or syntax mistakes. In addition, we have asked our colleagues who are skilled authors of English language papers to check the English quality of our manuscript. We believe that the language is now acceptable for the review process.

Reviewer 2 Report
This manuscript reports the effects of fertilizer (N, P) levels on rice phenotype, microclimate in rice paddy and blight sheath. In consideration of food security and global climate change, this topic is important, however, the study is not well presented by the current version of the manuscript.
I suggest the authors to clearify the following questions
1. summurize clear scientifc questions or hypothesis in the abstract;
2. introduce the background and purpose of this study, cite relative references in the introduction part, for example: Line38-39, there is no reference to support the statement;
3. tables and figures are not well explained, write more details about the listed results on foot note or figure caption;
4. the experimental design is not clear enough, split-plot design? duplications of each plot?
5. statistic analysis, what methods or models were used?
Author Response
Response to Reviewer 2
Comments and Suggestions for Authors
This manuscript reports the effects of fertilizer (N, P) levels on rice phenotype, microclimate in rice paddy and blight sheath. In consideration of food security and global climate change, this topic is important, however, the study is not well presented by the current version of the manuscript.
I suggest the authors to clarify the following questions
Thank you very much for your useful comments to improve our manuscript, we are thankful to the reviewer for their valuable time and proficient suggestions. Below is the point-by-point response to reviewer comments.
Summarize clear scientific questions or hypothesis in the abstract;
- Thank you for this suggestion, we have revised our abstract in the revised manuscript.
- Introduce the background and purpose of this study, cite relative references in the introduction part, for example: Line38-39, there is no reference to support the statement;
- Thank you for this suggestion. We added some references (Line 39-41).
- Tables and figures are not well explained, write more details about the listed results on foot note or figure caption;
- Thank you for this valuable comment. We revised and added more details in all Tables and figures footnotes.
- The experimental design is not clear enough, split-plot design? duplications of each plot?
- Thank you for this comment. We added the description of the field experiment site and (Lines 433-445 and 485-487).
- Statistical analysis, what methods or models were used?
- Thank you for this valuable comment. We have added the statistical analysis and models we used (Lines 484-487).

Round 2
Reviewer 2 Report
1. Double check the title, abstract and the whole manuscript, some expressions are not appropriate for this study, such as: "plant type", "severity"
2. Material and method part: it would be better to provide the longitude and latitude of the experimental site.
3. Why this cultivar was selected in this study, is it a wide planted cultivar in the local production area?
4. Material and method part: date for harvesting of the rice should be shown.
5. Material and method part: according to the manuscript, the heading stage and grain filling stage only last for 5 days, respectively. The duration is too short than normal rice cultivars, please clarify the reason for the short duration.
6. Material and method part: provide more information about "effective panicles", the definition and the measurement.